# ADAMTS13, VWF, and Endotoxin Are Interrelated and Associated with the Severity of Liver Cirrhosis via Hypercoagulability

**DOI:** 10.3390/jcm11071835

**Published:** 2022-03-26

**Authors:** Hiroaki Takaya, Tadashi Namisaki, Shohei Asada, Satoshi Iwai, Takahiro Kubo, Junya Suzuki, Masahide Enomoto, Yuki Tsuji, Yukihisa Fujinaga, Norihisa Nishimura, Yasuhiko Sawada, Kosuke Kaji, Hideto Kawaratani, Kei Moriya, Takemi Akahane, Masanori Matsumoto, Hitoshi Yoshiji

**Affiliations:** 1Department of Gastroenterology, Nara Medical University, Kashihara 634-8522, Japan; tadashin@naramed-u.ac.jp (T.N.); asahei@naramed-u.ac.jp (S.A.); satoshi181@naramed-u.ac.jp (S.I.); kubotaka@naramed-u.ac.jp (T.K.); suzukij@naramed-u.ac.jp (J.S.); masahidee@naramed-u.ac.jp (M.E.); tsujih@naramed-u.ac.jp (Y.T.); fujinaga@naramed-u.ac.jp (Y.F.); nishimuran@naramed-u.ac.jp (N.N.); yasuhiko@naramed-u.ac.jp (Y.S.); kajik@naramed-u.ac.jp (K.K.); kawara@naramed-u.ac.jp (H.K.); moriyak@naramed-u.ac.jp (K.M.); stakemi@naramed-u.ac.jp (T.A.); yoshijih@naramed-u.ac.jp (H.Y.); 2Department of Blood Transfusion Medicine, Nara Medical University, Kashihara 634-8522, Japan; mmatsumo@naramed-u.ac.jp

**Keywords:** ADAMTS13, von Willebrand factor, endotoxin, liver cirrhosis, hypercoagulability

## Abstract

ADAMTS13 specifically cleaves the multimeric von Willebrand factor (VWF), and an imbalance between ADAMTS13 activity (ADAMTS13:AC) and VWF antigen (VWF:Ag) levels is associated with the severity of liver cirrhosis (LC). However, the reason for this imbalance in patients with LC is unknown. This study investigated the relationship among ADAMTS13:AC, VWF:Ag, and endotoxin (Et) levels in patients with LC. ADAMTS13:AC and VWF:Ag levels were determined using ELISA, whereas Et levels were estimated using a chromogenic substrate assay. The levels of ADAMTS13 inhibitor (ADAMTS13:INH) were evaluated by measuring the extent that heat-inactivated patient’s plasma reduces the ADAMTS13:AC of the control. The status (degraded, normal, or unusually large [UL]) of the VWF multimer (VWFM) was determined through vertical agarose gel electrophoresis. ADAMTS13:AC, VWF:Ag, and Et levels decreased, increased, and increased, respectively, with the severity of LC. Patients with cirrhosis with high Et levels had lower and higher ADAMTS13:AC and VWF:Ag levels, respectively, than those with low Et levels. Patients with cirrhosis with detectable ADAMTS13:INH had higher Et levels than those with undetectable ADAMTS13:INH. Patients whose VWFM was either normal or UL had higher Et levels than those with degraded VWFM. In conclusion, ADAMTS13, VWF, and Et may be interrelated and associated with the severity of LC via hypercoagulability.

## 1. Introduction

Endotoxins (Et) are a component of the Gram-negative bacterial cell wall and are also known as lipopolysaccharides (LPS), whose core lipid A component comprises the toxic moiety [1]. Since endotoxemia was first associated with human liver diseases in the 1970s, much attention has been paid to the importance of intestinal Et as a critical cofactor in severe liver diseases [2,3]. Et are frequently detected in patients with liver cirrhosis (LC) who present with no other evidence of Gram-negative infection, indicating that their condition is due to the impaired hepatic clearance of gut-derived Et that are normally absorbed from the gastrointestinal tract [4,5]. Increased Et levels due to increased intestinal permeability and insufficient Et clearance by the hepatic reticuloendothelial system, including macrophages (e.g., Kupffer cells), may lead to a marked increase in liver sensitivity to Et, resulting in spillover of the Et into the systemic circulation, which leads to extrahepatic manifestations consistent with liver injury [1,2,3,4,5,6]. Increasing levels of Et in patients with LC are associated with hepatic failure, hepatic encephalopathy, and increased mortality [5,6,7]; they also trigger the hypercoagulability and cytokine cascade that precede multiple organ failure (MOF) (e.g., acute-on-chronic liver failure (ACLF)) [2,3,5].

A disintegrin-like and metalloproteinase with thrombospondin type-1 motifs 13 (ADAMTS13) is a metalloproteinase that specifically cleaves the multimeric von Willebrand factor (VWF) between Tyr1605 and Met1606 residues in the A2 domain [8,9,10,11,12,13,14,15]. VWF is a multimeric protein that performs two key functions in hemostasis. It is involved in the interaction between platelets and vascular endothelial cells (ECs) at the site of vascular damage and in the interaction among platelets. VWF also forms a complex with factor VIII, protecting it from proteolysis by activating protein C. The adhesive activity of VWF depends on the presence of high-molecular weight multimers [16]. The quantitative and/or qualitative abnormalities in the VWF adhesive plasma protein are the causes of the most common inherited bleeding disorder, von Willebrand disease [17]. In the absence of ADAMTS13, VWF multimers (VWFMs) are improperly cleaved, forming unusually large (UL) VWFMs that are released from vascular ECs. These UL VWFMs may then accumulate and, under conditions of high shear stress, induce the formation of platelet thrombi in the microvasculature [18,19]. It is currently thought that a severe deficiency in ADAMTS13 activity (ADAMTS13:AC) is a specific feature of thrombotic thrombocytopenic purpura (TTP) [20,21,22], which is a condition wherein platelet microthrombi are formed in systemic organs. We recently demonstrated that ADAMTS13 is produced exclusively in hepatic stellate cells (HSCs) located adjacent to ECs [18]. This location is also where VWF is produced and where ADAMTS13:AC is low in patients with advanced LC [23] compared to those with normal livers. HSCs are the major ADAMTS13-producing cells in the liver [18]; thus, previous studies have focused on the potential role of ADAMTS13 in the pathophysiology of liver diseases associated with sinusoidal and/or systemic microcirculatory disturbances [8,12,14,15,20,21,22,24,25,26,27]. ADAMTS13:AC and VWF antigen (VWF:Ag) levels were significantly lower and higher, respectively, in patients with alcoholic hepatitis [24,25,26,27], LC [20,21,22], ACLF [8], and acute liver failure [14,28] than in healthy subjects. Furthermore, ADAMTS13:AC may be a useful prognostic marker whose predictive performance is equal or superior to the Child–Pugh and Model For End-Stage Liver Disease scores, not only in the short-term, but also for the long-term survival of patients with LC [15].

Despite the usefulness of ADAMTS13:AC as a prognostic tool, the reason for the imbalance between the ADAMTS13 enzyme and VWF substrate in patients with LC is unknown. Previous studies have reported that in patients with sepsis, disseminated intravascular coagulation, and/or MOF, the levels of ADAMTS13:AC and VWF:Ag are lower and higher, respectively, than those of healthy subjects and that these levels are associated with higher levels of Et [12,14,29,30,31]. Furthermore, when healthy volunteers were given an intravenous infusion of Et, their levels of ADAMTS13:AC and VWF:Ag decreased and increased, respectively, compared to their previous levels [32].

Based on the abovementioned findings, it is possible that ADAMTS13, VWF, and Et are interrelated and associated with the severity of LC. Hence, we investigated the relationship(s) among ADAMTS13:AC, VWF:Ag, and parameters related to Et in patients with LC to determine the cause of the imbalance between ADAMTS13:AC and VWF:Ag in these patients.

## 2. Patients and Methods

### 2.1. Patients

A total of 99 patients with chronic liver diseases were included in this study (Figure 1). Of these, 15 had chronic hepatitis (CH) and 84 had LC. None of the included patients had any history of coagulopathy, sepsis, or platelet disorders. The following were the origins of the patients’ liver disease: the hepatitis B virus in 12 cases; hepatitis C virus in 67 cases; alcohol abuse in 10 cases; nonalcoholic steatohepatitis in 6 cases; primary biliary cirrhosis in 3 cases; and unknown in 1 case. LC was diagnosed through physical examinations, laboratory analyses, and imaging procedures as recommended by the 2020 evidence-based clinical practice guidelines for LC of the Japan Society of Gastroenterology and the Japan Society of Hepatology. In many cases, the diagnoses were confirmed by histological tests. Of the patients with LC, 25 were Child–Pugh class A, 23 were Child–Pugh class B, and 36 were Child–Pugh class C. All subjects gave their informed consent to participate in the study. This study was approved by the local Ethics Committee of Nara Medical University and was conducted in accordance with the ethical standards of the Declaration of Helsinki.

### 2.2. Determination of Endotoxin

All blood samples were obtained aseptically by peripheral venipuncture using pyrogen-free syringes and needles. Each blood sample was placed in a pyrogen-free tube and mixed with 1/10th volume of 3.8% sodium citrate as an anticoagulant. All tubes were placed on ice and transported immediately to the laboratory. Plasma was immediately separated in a refrigerated centrifuge at 3000× *g* at 4 °C for 15 min and stored at −80 °C for subsequent analysis. Et level was measured using a chromogenic substrate assay (Toxicolor LS–M Set, Seikagaku Kogyo Co., Tokyo, Japan) that implements kinetics analysis [33]. In brief, 50 μL of plasma was mixed with 450 μL of 0.02% Triton X-100, and the mixture was heated at 70 °C for 10 min to inactivate any inhibitor reacting with the Et. A standard curve was made using solutions with final exogenous Et concentrations of 180, 90, 45, 22.5, 11.3, and 5.6 pg/mL. Absorbance was measured at 37 ℃ every 15 s until 30 min using a microprocessor-controlled reader (Wellreader, SK603, Seikagaku Co., Tokyo, Japan). The linear part of the kinetics curve was read, and endogenous plasma Et levels were calculated from the standard curve. All measurements were done in duplicate, and the data are presented as mean values.

### 2.3. Determination of Levels of ADAMTS13:AC, VWF:Ag, and ADAMTS13:INH, and VWFM Pattern Analysis

Blood samples were taken from patients at the time of admission or during their hospital stay. Blood samples were stored in plastic tubes containing 1/10th the volume of 3.8% sodium citrate. Platelet-poor plasma was prepared by centrifuging plasma at 3000× *g* at 4 °C for 15 min and storing the plasma in aliquots at −80 °C until analysis. Plasma ADAMTS13:AC was determined using a sensitive chromogenic enzyme-linked immunosorbent assay (ELISA) kit (Kainos Laboratories Inc., Tokyo, Japan) [34]. ADAMTS13:AC levels among healthy subjects were 99% ± 22%. Plasma levels of VWF:Ag were measured using rabbit polyclonal sandwich ELISA (Dako, Glostrup, Denmark), and its level in healthy subjects was 102% ± 33% [35]. The levels of ADAMTS13 inhibitor (ADAMTS13:INH) were evaluated using plasma that had been heat-inactivated at 56°C for 30 min [14,20,24]. One Bethesda unit (BU) of inhibitor is defined as the amount of plasma that reduces ADAMTS13:AC to 50% of the control, and its titer was considered significant at >0.5 BU/mL [14,20,24]. VWFM patterns were analyzed by SDS–1.0% agarose electrophoresis followed by Western blotting with luminographic detection [14,20,24,36] and evaluated using image J (National Institutes of Health, Bethesda, MD, USA). VWFM patterns were classified as degraded, normal, or UL based on the increasing molecular weight of each type, respectively.

### 2.4. Statistical Analysis

Statistical analyses were performed using EZR (version 1.54, Saitama Medical Center, Jichi Medical University, Saitama, Japan), which is a graphical user interface for R (version 4.0.3, R Foundation for Statistical Computing, https://www.r-project.org (accessed on 17 February 2022)). EZR is a modified version of R commander version 2.7-1 that includes statistical functions frequently used in biostatistics [37]. Results are reported as median and interquartile ranges. The differences of between patients with cirrhosis based on Child–Pugh classification, ADAMTS13:AC, and its related parameters were first analyzed by the Kruskal–Wallis rank test, followed by Steel–Dwass test for groups showing significant differences. Categorical data were analyzed using Fisher’s exact test. Correlations were calculated by the Spearman rank test. A two-tailed *p*-value of less than 0.05 was considered significant.

## 3. Results

### 3.1. Clinical Characteristics of Patients

Clinical data of patients with CH and LC are provided in Table 1. Albumin levels (Alb), prothrombin time (PT), and platelet counts (Plt) reduced with increasing severity of LC, while total bilirubin (T-Bil) and blood urea nitrogen levels increased. Child–Pugh class C patients had the highest and lowest creatinine (Cre) and hemoglobin levels, respectively. Child–Pugh class C patients also had the highest tendency to suffer from ascites, hepatorenal syndrome (HRS), and encephalopathy. More patients with LC had esophageal varices and hepatocellular carcinoma (HCC) compared to patients with CH.

### 3.2. Plasma Levels of ADAMTS13:AC, VWF:Ag, and ADAMTS13:INH

ADAMTS13:AC levels decreased, while VWF:Ag levels increased as LC became more severe (Table 2). Consequently, the ratio of VWF:Ag to ADAMTS13:AC (VWF:Ag/ADAMTS13:AC) increased as LC progressed (Table 2). ADAMTS13:INH was detected in 1 (4%) Child–Pugh class A patient, 4 (17.4%) Child–Pugh class B patients, and 13 (36.1%) Child–Pugh class C patients (Table 2). ADAMTS13:AC was directly correlated (*p* < 0.05) with Alb (r = 0.405) (Figure 2a), PT (r = 0.423) (Figure 2b), and Plt (r = 0.302) (Figure 2c) but inversely correlated (*p* < 0.05) with T-Bil (r = −0.441) (Figure 2d) and Cre (r = −0.289) (Figure 2e).

### 3.3. Plasma Endotoxin Level and Its Relationship to ADAMTS13:AC, VWF:Ag, ADAMTS13:INH, and VWFM Patterns

Et levels increased as LC became more severe, and the highest levels were observed in Child–Pugh class C patients (Figure 3). Patients with Et levels > 5 pg/mL had lower ADAMTS13:AC (Figure 4a), while their VWF:Ag (Figure 4b) and VWF:Ag/ADAMTS13:AC (Figure 4c) levels were higher than those of patients whose Et levels were ≤ 5 pg/mL (*p* < 0.05). Et levels were inversely correlated with those of Alb (r = −0.307) (Figure 5a) and PT (r = −0.308) (Figure 5b), while they were directly correlated with T-Bil (r = 0.295) (Figure 5c) and Cre (r = 0.246) levels (Figure 5d) (*p* < 0.05). Patients with detectable ADAMTS13:INH had higher Et levels than those with undetectable ADAMTS13:INH (*p* < 0.05) (Figure 6a). Among the 34 patients whose ADAMTS13:AC was <50%, patients with normal and UL VWFM patterns had higher Et levels than those with degraded VWFM patterns (*p* < 0.05) (Figure 6b).

## 4. Discussion

The results of our study show that as cirrhosis progressed, ADAMTS13:AC levels gradually decreased, while VWF:Ag levels gradually increased. Moreover, Et levels also increased gradually as cirrhosis progressed, with the highest levels observed in Child–Pugh class C patients. Furthermore, patients with high Et levels had lower ADAMTS13:AC and higher VWF:Ag levels than those with low Et levels. These results indicate that ADAMTS13 and VWF are closely associated with Et in LC. In addition, previous work has established that Et produced by gut microbiota regulates the synthesis of VWF in vascular ECs via Toll-like receptor-2 [38]. Moreover, it is well-established that patients with LC also have gut dysbiosis [39] and that gut dysbiosis is associated with LC progression [2,3]. Low ADAMTS13:AC compared with VWF:Ag levels (i.e., the imbalance between ADAMTS13 enzyme and VWF substrate) may be associated with the severity of LC through gut dysbiosis and an increase in Et levels.

Previous work has shown that a decrease in ADAMTS13:AC and increase in VWF:Ag levels are associated with hepatic encephalopathy, HRS, and ascites [20,21]. Moreover, Et promote the synthesis of endothelin and nitric oxide in HSCs and ECs, which results in increased portal blood pressure [40]. Because the hemodynamic effects of portal hypertension (PHT) can induce hepatic encephalopathy [41], HRS [42], and ascites [43], it is possible that ADAMTS13:AC and VWF:Ag are associated with these conditions through Et-induced development of hepatic encephalopathy, HRS, ascites, and subsequent PHT.

The pathophysiology of acquired TTP is associated with acquired autoantibodies against ADAMTS13 (i.e., ADAMTS13:INH) [44]. We found that 21.4% of the patients with LC had detectable levels of ADAMTS13:INH, and the proportion of the patients with detectable levels of ADAMTS13:INH increased as LC progressed. Among healthy volunteers, the intravenous infusion of Et results in higher VWF:Ag and lower ADAMTS13:AC levels than those in volunteers who received no Et infusion [32]. Thus, Et may act directly to decrease ADAMTS13:AC and increased VWF:Ag levels. In our study, patients with detectable ADAMTS13:INH had higher Et levels than those with undetectable ADAMTS13:INH. We therefore speculate that Et is an ADAMTS13:INH.

A previous study has reported that in patients with end-stage cirrhosis, a marked decrease in ADAMTS13:AC (i.e., <3%) is associated with an IgG type of ADAMTS13:INH [20]. These patients meet the diagnostic criteria for TTP, i.e., the formation of platelet microthrombi in systemic organs, although these patients show no apparent clinical features of TTP. These patients may have a condition similar to TTP, or they may have subclinical TTP. In addition, a previous study has reported that patients with venous thromboembolism (VTE) have lower ADAMTS13:AC and higher VWF:Ag than those without VTE, indicating that the imbalance between ADAMTS13 enzyme and VWF substrate is associated with VTE [45]. Existing evidence also suggests that the imbalance between ADAMTS13 enzyme and VWF substrate is associated with a high-risk state of platelet microthrombi formation, which may be closely related to functional hepatic reserve.

Platelet microthrombi formation contributes to PHT through multiple pathways [46]. Therefore, the development of hepatic encephalopathy, HRS and ascites may be associated with platelet microthrombi formation because the development of these disorders is associated with PHT. We speculate that the development of hepatic encephalopathy, HRS, and ascites is associated with platelet microthrombi formation through an imbalance between ADAMTS13 enzyme and VWF substrate.

The present and some previous studies reported UL VWFM in several patients with LC [20,21,22]. VWFM patterns reflect hypercoagulability, i.e., the tendency for VWFM to coagulate is in the order degraded < normal < UL. Moreover, the imbalance between ADAMTS13 enzyme and VWF substrate also reflects hypercoagulability. In our study, patients with normal and UL VWFM patterns had higher Et levels than those with a degraded VWFM pattern. In other words, platelet microthrombi formation was associated with both VWFM pattern and the imbalance between ADAMTS13 enzyme and VWF substrate, which were affected by an increase in Et levels.

We previously showed that ADAMTS13:AC may be a useful prognostic marker in patients with LC [15] because ADAMTS13:AC is associated with functional hepatic reserve. It is widely known that kidney function is related to the prognosis of patients with LC [47], and in this study, we show that ADAMTS13:AC and Et levels are associated with Cre levels. We have previously reported that ADAMTS13:AC is associated with HRS [20,21]; therefore, ADAMTS13:AC may be associated with prognosis of LC through both kidney function and functional hepatic reserve. In addition, Et may affect the association between ADAMTS13:AC and kidney function.

Et may be more closely associated with the pathophysiology of LC than the imbalance between ADAMTS13 enzyme and VWF substrate because it is directly associated with LC progression [1,2,3,4,5,6]. Nevertheless, the present and previous evidence suggest that as LC progresses, ADAMTS13:AC and VWF:Ag levels change more gradually than Et levels and that IgG type of ADAMTS13:INH that meets the diagnostic criteria for TTP was detected in some patients with end-stage cirrhosis; especially a marked decrease in ADAMTS13:AC (i.e., <3%) [20]. Additionally, Plt levels usually decrease as LC progresses. In the present study, ADAMTS13:AC, but not Et levels, was directly correlated with Plt levels. Based on our results, we believe that the imbalance between ADAMTS13 enzyme and VWF substrate is directly associated with the pathophysiology of LC via hypercoagulability, and that Et-induced LC progression was also directly involved in liver injury. Thus, LC progression is involved in liver injury via the increase in imbalance between ADAMTS13 enzyme and VWF substrate. Further studies are required to investigate the relationship between ADAMTS13 enzyme–VWF substrate imbalance and Et in patients with LC.

The present study has several limitations. Our sample size was small, and secondly, all enrollees were from a single study center. Additionally, the thrombosis, inflammation, and platelet disorders may affect the ADAMTS13:AC, VWF:Ag, and Et levels in patients with LC [12,45,48]. None of the patients included in the present study had any history of coagulopathy, sepsis, or platelet disorders. We previously reported that the imbalance between ADAMTS13 enzyme and VWF substrate was associated with HCC [10,13]. Therefore, there was a possibility that HCC influenced the results of the present study. However, as there was no difference in the proportion of the patients with HCC among Child–Pugh classes A, B, and C, we believe that HCC did not affect the results of the present study. In the present study, 67.7% of all patients were infected with the hepatitis C virus. Some previous studies reported that interferon and direct acting antivirals (DAAs) helped in recovery from the imbalance between ADAMTS13 enzyme and VWF substrate and high Et levels in patients with LC and hepatitis C [8,49,50]. In the present study, we could not investigate the history of interferon or DAA treatment. However, we believe that the history of interferon or DAA treatment did not influence the results of the present study because almost all parameters, including ADAMTS13:AC, VWF:Ag, Et levels, functional hepatic reserve, and kidney function, in patients with hepatitis C recovered following interferon or DAA treatment [8,49,50,51,52]. In summary, the results of our study indicate that ADAMTS13: AC, VWF: Ag, and Et are interrelated and associated with the severity of LC via hypercoagulability.

## Figures and Tables

**Figure 1 jcm-11-01835-f001:**
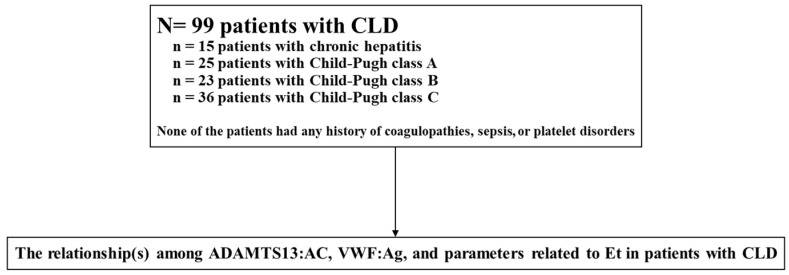
Study design. A total of 99 patients with CLD were included in this study. None of the included patients had any history of coagulopathy, sepsis, or platelet disorders. We investigated the relationship(s) among ADAMTS13:AC, VWF:Ag, and parameters related to Et in patients with CLD to determine the cause of the imbalance between ADAMTS13:AC and VWF:Ag in these patients. CLD, chronic liver diseases; ADAMTS13, a disintegrin-like and metalloproteinase with thrombospondin type 1 motifs 13; ADAMTS13:AC, ADAMTS13 activity; VWF, von Willebrand factor; VWF:Ag, VWF antigen; Et, endotoxins.

**Figure 2 jcm-11-01835-f002:**
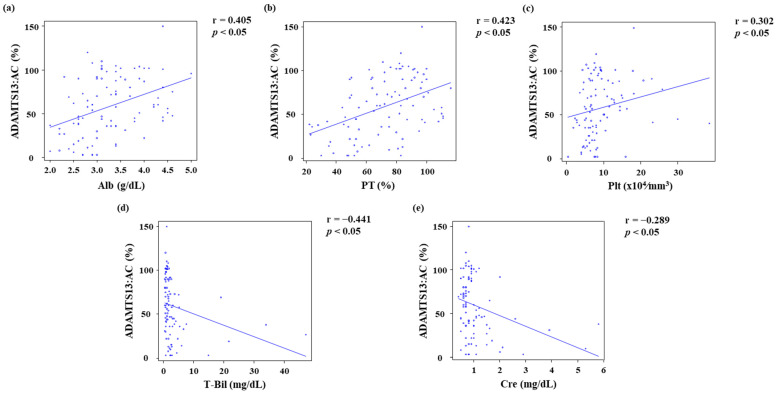
Relationship between ADAMTS13:AC and participant characteristics. (**a**–**c**) ADAMTS13:AC directly correlated with Alb (r = 0.405, *p* < 0.05), PT (r = 0.423, *p* < 0.05), and Plt (r = 0.302, *p* < 0.05). (**d**,**e**) ADAMTS13:AC inversely correlated with T-Bil (r = −0.441, *p* < 0.05) and Cre (r = −0.289, *p* < 0.05). ADAMTS13, a disintegrin-like and metalloproteinase with thrombospondin type 1 motifs 13. ADAMTS13:AC, ADAMTS13 activity; Alb, albumin; PT, prothrombin time; Plt, platelet count; T-Bil, total bilirubin; Cre, creatinine.

**Figure 3 jcm-11-01835-f003:**
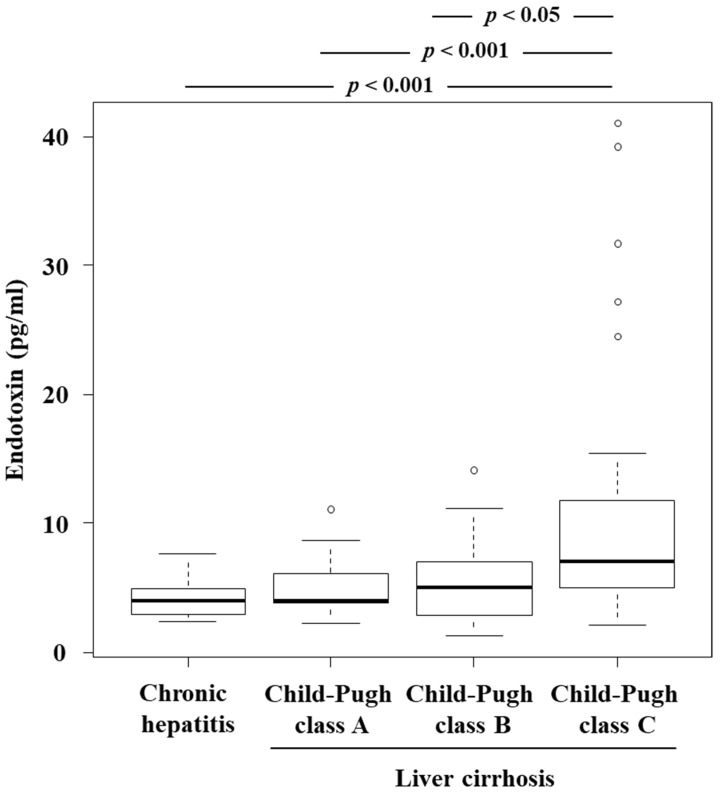
Relationship between endotoxin levels and liver cirrhosis. Endotoxin levels increased as LC progressed and were highest in patients with Child–Pugh class C cirrhosis (*p* < 0.001 for between Child–Pugh classes C and A or chronic hepatitis, and *p* < 0.05 for between Child–Pugh classes C and B). Box plots represent, from top to bottom, sample maximum, upper-quartile range, median and lower-quartile range, and sample minimum. LC, liver cirrhosis.

**Figure 4 jcm-11-01835-f004:**
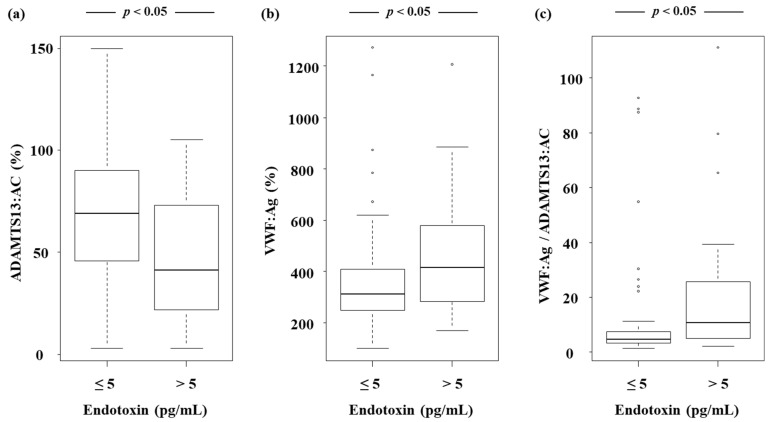
Relationship between endotoxin levels and ADAMTS13:AC, endotoxin levels and VWF:Ag, and endotoxin levels and VWF:Ag/ADAMTS13:AC. (**a**) Patients with endotoxin levels of >5 pg/mL had lower ADAMTS13:AC than those with endotoxin levels of ≤5 pg/mL (*p* < 0.05). (**b**) Patients with endotoxin levels of >5 pg/mL had higher VWF:Ag than those with endotoxin levels of ≤5 pg/mL (*p* < 0.05). (**c**) Patients with endotoxin levels of >5 pg/mL had higher VWF:Ag/ADAMTS13:AC than those with endotoxin levels of ≤5 pg/mL (*p* < 0.05). Box plots represent, from top to bottom, sample maximum, upper-quartile range, median and lower-quartile range, and sample minimum. ADAMTS13, a disintegrin-like and metalloproteinase with thrombospondin type 1 motifs 13; ADAMTS13:AC, ADAMTS13 activity; VWF, von Willebrand factor; VWF:Ag, VWF antigen; VWF:Ag/ADAMTS13:AC, ratio of VWF:Ag to ADAMTS13:AC.

**Figure 5 jcm-11-01835-f005:**
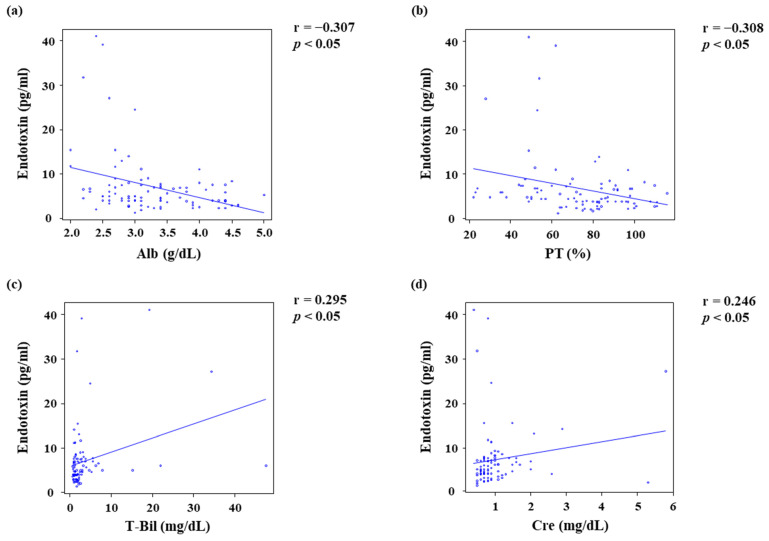
Relationship between endotoxin levels and patient characteristics. (**a**,**b**) Endotoxin levels were inversely correlated with Alb (r = −0.307, *p* < 0.05) and PT (r = −0.308, *p* < 0.05). (**c**,**d**) Endotoxin levels were directly correlated with T-Bil (r = 0.295, *p* < 0.05) and Cre (r = 0.246, *p* < 0.05). Alb, albumin; PT, prothrombin time; T-Bil, total bilirubin; Cre, creatinine.

**Figure 6 jcm-11-01835-f006:**
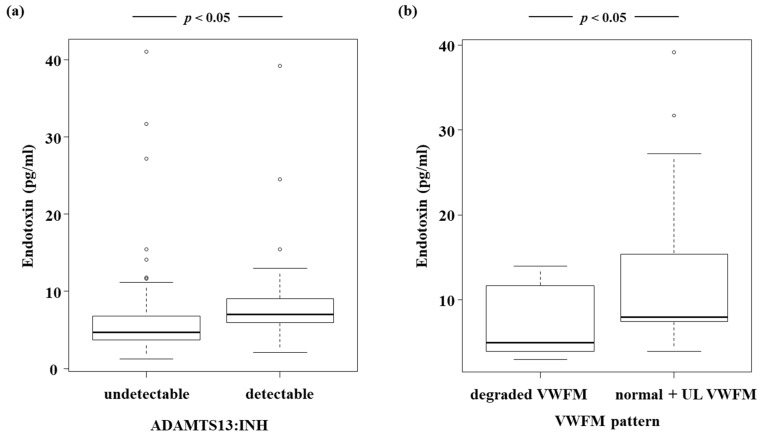
Relationship between endotoxin levels and ADAMTS13:INH and between endotoxin levels and VWFM patterns. (**a**) Patients with detectable ADAMTS13:INH had higher endotoxin levels than those with undetectable ADAMTS13:INH (*p* < 0.05). (**b**) Patients with normal and UL VWFM patterns had higher endotoxin levels than those with degraded VWFM (*p* < 0.05). Box plots represent, from top to bottom, sample maximum, upper-quartile range, median and lower-quartile range, and sample minimum. ADAMTS13, a disintegrin-like and metalloproteinase with thrombospondin type 1 motifs 13; ADAMTS13:INH, ADAMTS13 inhibitor; VWF, von Willebrand factor; VWFM, VWF multimer; UL, unusually large.

**Table 1 jcm-11-01835-t001:** Characteristics of patients with liver cirrhosis.

Variable	Total (*n* = 99)	Chronic Hepatitis (*n* = 15)	Liver Cirrhosis		
			Child–Pugh Class A (*n* = 25)	Child–Pugh Class B (*n* = 23)	Child–Pugh Class C (*n* = 36)
Age (year)	67 (60–74)	60 (51–67)	72 (68–76) *	64 (58–68)	69 (61–74) *
Sex (male/female)	63/36	9/6	19/6	14/9	21/15
Etiology (HBV/HCV/Alcohol/NASH/PBC/unknown)	12/67/10/6/3/1	0/15/0/0/0/0	1/18/4/2/0/0	6/15/1/1/0/0	5/19/5/3/3/1
Albumin (g/dL)	3.2 (2.8–3.8)	4.2 (3.9–4.5)	3.4 (3.1–4.1) *	3.2 (2.9–3.4) **	2.7 (2.5–3.0) ***
Aspartate aminotransferase (IU/L)	52 (38–76)	39 (31–48)	52 (38–64)	56 (41–89)	63 (38–85)
Alanine aminotransferase (IU/L)	42 (26–58)	50 (35–61)	38 (27–55)	45 (30–60)	32 (23–53)
Prothrombin time (%)	75 (53–89)	98 (94–105)	83 (80–92) *	71 (64–84) **	50 (38–57) ***
Blood urea nitrogen (mg/dL)	17 (12–26)	13 (10–15)	16 (10–20)	16 (14–22) *	26 (16–45) ***
Creatinine (mg/dL)	0.8 (0.7–1.1)	0.8 (0.6–0.8)	0.8 (0.7–0.9)	0.7 (0.6–1.0)	1.0 (0.7–1.5) *^,^ **
Total bilirubin (mg/dL)	1.6 (0.8–2.6)	0.6 (0.5–1.4)	1.0 (0.7–1.3) *	1.6 (0.9–1.9) **	3.4 (2.3–5.3) ***
White blood cell count (/mm^3^)	4400 (2975–6200)	5200 (3700–6150)	4400 (2900–5500)	3750 (2575–5100)	5250 (3300–8175)
Hemoglobin (g/dL)	11.4 (9.6–12.8)	14.0 (12.1–15.5)	12.0 (11.1–12.8) *	12.2 (10.9–13.0)	9.0 (7.7–10.9) ***
Platelet count (×10^4^/mm^3^)	7.7 (5.9–11.0)	14.2 (9.7–17.2)	9.2 (7.6–12.7) *	6.4 (5.2–8.3) **	6.4 (4.8–7.7) **
Ascites (none/mild to moderate/severe)	62/12/25	15/0/0	25/0/0	14/7/2 **	8/5/23 ***
Hepatorenal syndrome (with/without)	9/90	0/15	0/25	1/22	8/28 ***
Encephalopathy (with/without)	37/62	0/15	0/25	7/16 **	30/6 ***
Esophageal varices (with/without)	65/34	0/15	16/9 *	19/4 *	30/6 *
Hepatocellular carcinoma (with/without)	46/53	1/14	17/8 *	12/11 *	16/20 *

* *p* < 0.05 between chronic hepatitis and Child–Pugh class A, B, or C; ** *p* < 0.05 between Child–Pugh classes A and B or C; *** *p* < 0.05 between Child–Pugh classes B and C; data are expressed as medians (interquartile ranges); HBV, hepatitis B virus; HCV, hepatitis C virus; NASH, nonalcoholic steatohepatitis; PBC, primary biliary cholangitis.

**Table 2 jcm-11-01835-t002:** ADAMTS13 activity and associated parameters.

Variable	Total (*n* = 99)	Chronic Hepatitis (*n* = 15)	Liver Cirrhosis		
			Child–Pugh Class A (*n* = 25)	Child–Pugh Class B (*n* = 23)	Child–Pugh Class C (*n* = 36)
ADAMTS13:AC (%)	58 (35–89)	80 (59–93)	80 (60–102)	60 (36–92) **	27 (13–45) ***
VWF:Ag	341 (259–500)	290 (152–329)	315 (221–408) *	400 (310–626) **	415 (303–594)
VWF:Ag /ADAMTS13:AC	6.5 (3.5–16.6)	3.3 (2.5–4.6)	4.2 (3.1–5.5) *	7.2 (3.5–13.9) **	18.2 (10.3–33.2) ***
ADAMTS13:INH (with/without)	18/81	0/15	1/24	4/19	13/23

* *p* < 0.05 between chronic hepatitis and Child–Pugh class A; ** *p* < 0.05 between Child–Pugh classes A and B; *** *p* < 0.05 between Child–Pugh classes B and C; Data are expressed as medians (interquartile ranges); ADAMTS13, a disintegrin-like and metalloproteinase with thrombospondin type 1 motifs 13; ADAMTS13:AC, ADAMTS13 activity; VWF, von Willebrand factor; VWF:Ag, VWF antigen; VWF:Ag/ADAMTS13:AC, ratio of VWF:Ag to ADAMTS13:AC; ADAMTS13:INH, ADAMTS13 inhibitor.

## Data Availability

Informed consent for data sharing was not obtained but the presented data are anonymized, and the risk of identification is low.

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
