# Peer review of "ADAMTS13, VWF, and Endotoxin Are Interrelated and Associated with the Severity of Liver Cirrhosis via Hypercoagulability"

_jcm, 2022, doi:10.3390/jcm11071835_

Round 1
Reviewer 1 Report
The authors of the manuscript focused on ADAMTS13, VWF, and endotoxin in liver cirrhosis. Endothelial dysfunction is involved in the pathogenesis of portal hypertension and in the progression of liver disease. As an indicator of endothelial dysfunction, von Willebrand factor (vWF-Ag) can be a useful mortality predictor in patients with liver cirrhosis The authors have come up with an interesting articler that is very important for medical research. Higher von Willebrand factor antigen is the prominent hemostatic disorders in patients with cirrhosis. The cleaving protease ADAMTS13 regulates the multimeric structure of vWF. Cirrhotic patients have reduced ADAMTS13 levels with elevated amounts of high-molecular-weight vWF in circulationSome parts of the manuscript need to be corrected and supplemented in order for this manuscript to be published.
The manuscript is well structured, but some parts are missing some important facts that authors should add.
Page 2:
In the introduction, the authors should state the characteristics and function of the von Willebrand factor. VWF is a multimeric protein that performs two key functions in hemostasis. It is involved in the interaction between platelets and the subendothelium at the site of vascular damage, and in the interaction between platelets. VWF also forms a complex with factor VIII, protecting it from proteolysis by activating protein C. The adhesive activity of VWF depends on the presence of high-molecular weight (HMW) multimers. Authors should cite the manuscript in which it was publishedThis manuscript should be quoted by the authors.: Diagnostics 2021, 11(11), 2153; https://doi.org/10.3390/diagnostics11112153
Page 2:
In this section, the authors should mention that quantitative and / or qualitative abnormalities in the VWF adhesive plasma protein is the cause of the most common inherited bleeding disorder: von Willebrand disease. Authors should add a reference in which this data has been published. Semin Thromb Hemost 2017; 43(06): 639-64, DOI: 10.1055/s-0037-1603362
Tables and figures in the text are very clearly written.
I have to say that with these 46 references. The smaller half are references of published manuscripts for the last 5 years.
Author Response
We thank the reviewers for their careful review of our manuscript and for their useful comments. We have carefully revised the manuscript according to the comments, and our manuscript has been proofread once again. Our responses to the each of the reviewers’ comments are as follows:
Response to Reviewer 1
Thank you for reviewing our manuscript. Our responses to your comments are as follows:
1) In the introduction, the authors should state the characteristics and function of the von Willebrand factor. VWF is a multimeric protein that performs two key functions in hemostasis. It is involved in the interaction between platelets and the subendothelium at the site of vascular damage, and in the interaction between platelets. VWF also forms a complex with factor VIII, protecting it from proteolysis by activating protein C. The adhesive activity of VWF depends on the presence of high-molecular weight (HMW) multimers. Authors should cite the manuscript in which it was publishedThis manuscript should be quoted by the authors.: Diagnostics 2021, 11(11), 2153; https://doi.org/10.3390/diagnostics11112153
2) In this section, the authors should mention that quantitative and / or qualitative abnormalities in the VWF adhesive plasma protein is the cause of the most common inherited bleeding disorder: von Willebrand disease. Authors should add a reference in which this data has been published. Semin Thromb Hemost 2017; 43(06): 639-64, DOI: 10.1055/s-0037-1603362
Response to 1) and 2)
Thank you for your valuable comments. We have indicated the characteristics and functions of the von Willebrand factor and disease in the Introduction section as follows:
“VWF is a multimeric protein that performs two key functions in hemostasis. It is involved in the interaction between platelets and vascular endothelial cells (ECs) at the site of vascular damage and in the interaction among platelets. VWF also forms a complex with factor VIII, protecting it from proteolysis by activating protein C. The adhesive activity of VWF depends on the presence of high-molecular weight multimers (16). The quantitative and/or qualitative abnormalities in the VWF adhesive plasma protein are the cause of the most common inherited bleeding disorder, von Willebrand disease (17).”
Further, we added references (16) and (17).
Reviewer 2 Report
This is a nice paper regarding alterations of hemostasis in patients with cirrhosis. Some comments and suggestion:
1) HCC has been recently associated with specific alterations of hemostasis in cirrhosis. Were these patients included? If so, it may be worth to include and discuss these data.
2) Please include the flow-chart (including patients who were screened and considered not eligible and why).
3) Etiology of liver disease may also affect hemostasis. In this study, the vast majority of patients had HCV cirrhosis. Treatment of HCV is associated with significant modifications in hemostasis . Were these patients previously treated? It may be worth to include these data and perform a sub-analysis according to disease etiology.
4) The paper is OK. However, English proofreading may improve flow and synthaxis.
Author Response
We thank the reviewers for their careful review of our manuscript and for their useful comments. We have carefully revised the manuscript according to the comments, and our manuscript has been proofread once again. Our responses to the each of the reviewers’ comments are as follows:
Response to Reviewer 2
Thank you for reviewing our manuscript. Our responses to your comments are as follows:
1) HCC has been recently associated with specific alterations of hemostasis in cirrhosis. Were these patients included? If so, it may be worth to include and discuss these data.
Response to 1)
Thank you for your valuable comment. We have indicated the proportion of patients with HCC among Child–Pugh classes A, B, and C and whether HCC influenced the findings of the present study or not in the Results and Discussion sections as follows:
“More patients with LC had esophageal varices and hepatocellular carcinoma (HCC) compared to patients with CH.”
“We previously reported that the imbalance between ADAMTS13 enzyme and VWF substrate was associated with HCC (10, 13). Therefore, there was a possibility that HCC influenced the results of the present study. However, as there was no difference in the proportion of the patients with HCC among Child–Pugh classes A, B, and C, we believe that HCC did not affect the results of the present study.”
2) Please include the flow-chart (including patients who were screened and considered not eligible and why).
Response to 2)
Thank you for your valuable comment. We have added the flow-chart as Figure 1. All screened patients were included in the study.
3) Etiology of liver disease may also affect hemostasis. In this study, the vast majority of patients had HCV cirrhosis. Treatment of HCV is associated with significant modifications in hemostasis . Were these patients previously treated? It may be worth to include these data and perform a sub-analysis according to disease etiology.
Response to 3)
Thank you for your valuable comment. We have reported the relationship of the imbalance between ADAMTS13 enzyme and VWF substrate and high Et levels and the history of interferon or direct acting antiviral (DAA) treatment in the Discussion as follows:
“In the present study, 67.7% of all patients were infected with the hepatitis C virus. Some previous studies reported that interferon and direct acting antivirals (DAAs) helped in recovery from the imbalance between ADAMTS13 enzyme and VWF substrate and high Et levels in patients with LC and hepatitis C (8, 49, 50). In the present study, we could not investigate the history of interferon or DAA treatment. However, we believe that the history of interferon or DAA treatment did not influence the results of the present study because almost all parameters, including ADAMTS13:AC, VWF:Ag, Et levels, functional hepatic reserve, and kidney function, in patients with hepatitis C recovered following interferon or DAA treatment (8, 49-52).”
4) The paper is OK. However, English proofreading may improve flow and synthaxis.
Response to 4)
Thank you for your valuable comment. Our manuscript has undergone English proofreading once again.
Response to 3)
Thank you for your valuable comment. We have reported the relationship of the imbalance between ADAMTS13 enzyme and VWF substrate and high Et levels and the history of interferon or direct acting antiviral (DAA) treatment.
“In the present study, 67.7% of all patients were infected with the hepatitis C virus. Some previous studies reported that interferon and direct acting antivirals (DAAs) helped in recovery from the imbalance between ADAMTS13 enzyme and VWF substrate and high Et levels in patients with LC and hepatitis C (8, 49, 50). In the present study, we could not investigate the history of interferon or DAA treatment. However, we believe that the history of interferon or DAA treatment did not influence the results of the present study because almost all parameters, including ADAMTS13:AC, VWF:Ag, Et levels, functional hepatic reserve, and kidney function, in patients with hepatitis C recovered following interferon or DAA treatment (8, 49-52).”
4) The paper is OK. However, English proofreading may improve flow and synthaxis.
Response to 4)
Thank you for your valuable comment. Our manuscript has undergone English proofreading once again.
Round 2
Reviewer 1 Report
The presented manuscript has been corrected in response to the suggestions. The authors have followed the recommendations of the reviewer. After the revision, the provided data and addition of the results became more clear. I would like to thank the authors for resubmitting the manuscript and explaining the obscure points from the previous version.